# Altered paracrine signaling from the injured knee joint impairs postnatal long bone growth

Alberto Roselló-Díez[1]*, Daniel Stephen[1], Alexandra L Joyner[1,2]*

[1]Developmental Biology Program, Sloan Kettering Institute, New York, United States; [2]Biochemistry, Cell and Molecular Biology Program, Weill Cornell Graduate Schoolof Medical Sciences, New York, United States

**Abstract** Regulation of organ growth is a poorly understood process. In the long bones, the growth plates (GPs) drive elongation by generating a scaffold progressively replaced by bone. Although studies have focused on intrinsic GP regulation, classic and recent experiments suggest that local signals also modulate GP function. We devised a genetic mouse model to study extrinsic long bone growth modulation, in which injury is specifically induced in the left hindlimb, such that the right hindlimb serves as an internal control. Remarkably, when only mesenchyme cells surrounding postnatal GPs were killed, left bone growth was nevertheless reduced. GP signaling was impaired by altered paracrine signals from the knee joint, including activation of the injury response and, in neonates, dampened IGF1 production. Importantly, only the combined prevention of both responses rescued neonatal growth. Thus, we identified signals from the knee joint that modulate bone growth and could underlie establishment of body proportions.

*For correspondence: roselloa@mskcc.org (AR-D); joynera@mskcc.org (ALJ)

**Competing interests:** The authors declare that no competing interests exist.

## Introduction

The signaling pathways that underlie the control of organ size during development, and recovery from injury, have important implications for the treatment of growth disorders. The same pathways likely also underlie evolutionary differences in body sizes and proportions (*Haldane, 1926*). Inter-species transplantation experiments have shown that growth of most organs, including the limbs, eyes (*Twitty and Schwind, 1931*) and jaws (*Schneider, 2015*), to a great extent follows an intrinsic genetic program, as the final size of the organ graft is close to that of the donor. However, extrinsic factors such as nutrient availability, inflammation, mechanical forces or the presence of additional organ grafts can modulate, to varying extents, growth of many organs (reviewed in [*Roselló-Díez and Joyner, 2015*]). Therefore, interactions between organ-intrinsic and extrinsic cues must be critical to the regulation of individual organ size, relative body proportions and responses to developmental insults (*Stanger, 2008*; *Parker, 2011*; *Mirth and Shingleton, 2012*).

The developing vertebrate limb is an excellent model to study pathways controlling organ growth, as it is amenable to non-lethal manipulation, and it is composed of multiple tissues (muscles, tendons, bone, nerves, vessels, etc.) that grow coordinately and are exposed to biochemical and mechanical extrinsic signals that interact with their intrinsic genetic growth programs. Long bones within the limbs have been popular models for studies of patterning, growth and evolution since at least the nineteenth century (*Owen, 1849*), mostly as isolated entities, although lately also as part of the musculo-skeletal unit (*Berendsen and Olsen, 2015*; *Shwartz et al., 2013*). Long bone growth occurs through endochondral ossification, driven by a transient cartilage structure, the growth plate (GP), located at both ends (epiphyses) of the bone (reviewed in [*Kronenberg, 2003*]). Once mesenchymal cells condense into the anlage of the skeletal elements, they differentiate into collagen II-

**eLife digest** As bones grow, their size is carefully controlled and coordinated with the growth of the other organs in the body. The mechanisms that control organ size also help the body to recover from injury, and play a key role in controlling body size and proportions. Over the course of evolution, these mechanisms have likely changed to produce the distinct body sizes and proportions seen in humans and other animals.

Despite their importance, it is not well understood how signals from both inside and outside an organ work together to regulate its size. In growth disorders this signaling goes wrong, which can lead to a person having unusual proportions such as a very short stature or having one leg shorter than the other. Currently, most growth disorders that affect leg proportions are treated with painful surgical procedures. Researchers would like to know how bone growth is affected by signals from the surrounding tissues because this could help them to develop new non-invasive treatments for these conditions.

Long bones, for example those in the leg, grow from structures near their ends called growth plates. Roselló-Díez et al. have now engineered mice in which an injury shortly after birth caused cells in the knee in the rear left leg to die off. At the same time, the rear right leg of the mice developed as normal, allowing the growth of the two legs to be compared. Roselló-Díez et al. found that the left leg of these mice grew more slowly than the right leg, even though none of the cells in the growth plate of the left leg bone had been damaged. Further investigation revealed that this was because the injury caused an imbalance between the growth-promoting and growth-restricting signals that are produced by the fat pad and articular cartilage in the knee joint. Restoring the lost balance allowed the left leg bone to grow to a more normal length.

In the future, boosting bone growth signals might provide a way to treat conditions like dwarfism or leg-length discrepancies. Understanding how different tissues influence body proportions could also help researchers to investigate how different animals evolved different body proportions.

expressing chondrocytes that undergo sequential differentiation from bone ends to center. Resting (quiescent) chondrocytes produce proliferative cells that after a few rounds of duplication cease proliferation and start to differentiate into hypertrophic chondrocytes (HTCs). HTCs increase their volume as they progress towards the center of the bone, laying down a collagen X-rich extracellular matrix (ECM) and secreting factors that recruit vasculature and bone precursors (osteoblasts) into the skeletal element. Some HTCs die by apoptosis, while others transdifferentiate into osteoblasts (*Yang et al., 2014*; *Zhou et al., 2014*), which also derive from a fibrous layer that wraps the cartilage (perichondrium) (*Maes et al., 2010*; *Kronenberg, 2007*), and both osteoblast pools form the primary ossification center by replacing the cartilaginous ECM with bone. This invasion of osteoblasts, blood vessels and ossification process is later recapitulated in the center of both epiphyseal regions, giving rise to the secondary ossification centers. The GP remains as a cartilage disc between the primary and secondary ossification centers, and responds to both intrinsic and extrinsic factors that regulate bone length. Within the GP, a negative feedback loop between indian hedgehog (IHH), secreted by pre-hypertrophic chondrocytes, and parathyroid hormone-like peptide (PTHLH or PTHrP) secreted by resting chondrocytes, couples chondrocyte proliferation and differentiation (*Karp et al., 2000*; *Vortkamp et al., 1996*; *Lee et al., 1996*; *Long et al., 2001*), and is the main conduit through which other local signals, such as fibroblast growth factors (FGFs) and bone morphogenetic proteins (BMPs) exert their function (*Roselló-Díez and Joyner, 2015*). A number of systemic signals (hormones) have long been known to impact on bone growth (*Roselló-Díez and Joyner, 2015*), but the contribution of local signals extrinsic to the GP has been explored less. Interestingly, the GPs at each end of the long bones contribute differentially to final bone size (*Digby, 1916*; *Payton, 1932*; *Moss-Salentijn, 1974*), raising the possibility that each has a different intrinsic growth potential, and/or that each is exposed to distinct local environmental cues. Indeed, while there has been an emphasis on intrinsic regulators of bone growth potential (*Nilsson and Baron, 2004*), classic GP transplantation experiments revealed that the local environment can modify the amount of growth produced by a GP (*Moss-Salentijn, 1974*; *Röhlig, 1969*; *Hert, 1964*). The

identities of such local extrinsic signal(s), however, have only recently begun to emerge. For example, TGFβR2 signaling from the interzone (the embryonic precursor of the joint [*Pacifici et al., 2006*]) affects maturation and signaling in the hypertrophic zone (HZ) (*Longobardi et al., 2012*); while some human variants located near the gene *GDF5*, expressed in the interzone, are associated with decreased height and increased osteoarthritic risk (*Sanna et al., 2008*; *Wu et al., 2012*). Another important player is WNT signaling from the interzone, as its balance with BMP signaling from the GP controls the differentiation of chondrogenic precursors towards either GP (transient cartilage) or articular (permanent cartilage) fate (*Ray et al., 2015*). In addition, a well-known modulator of bone growth is the Growth Hormone/Insulin-like growth factor (GH/IGF) axis, whereby GH induces *Igf1* expression in several tissues, including the liver and the GP, but also plays an IGF1-independent role in postnatal growth (*Isgaard et al., 1988*; *Lupu et al., 2001*). Importantly, the accumulated evidence suggests that local rather than systemic (i.e. liver-derived) IGF1 is the main factor controlling appendicular (limb) skeletal growth at early postnatal stages. For example, while genetic deletion of *Igf1* in the whole embryo affects skeletal growth from late gestation onwards (*Baker et al., 1993*), mice mutant for *growth hormone receptor* (*Ghr*) do not show a body weight or a limb length phenotype until postnatal day (P) 10–15 (*Lupu et al., 2001*; *Zhou et al., 1997*), and GH-induced, liver-specific IGF1 only begins exerting postnatal hormonal action shortly after 3 weeks of age in mice (*Stratikopoulos et al., 2008*). Similarly, deletion of *Igf1* in the liver (which diminishes circulating IGF1 levels by >75%) does not affect appendicular skeletal growth (*Sjögren et al., 1999*; *Yakar et al., 1999*), whereas chronic overexpression of *Igf1* in liver, brain and other organs (leading to increased serum IGF1 levels) results in overgrowth of only a subset of organs that does not include the skeleton (*Mathews et al., 1988*). Regarding local IGF1 sources, the expression of *Igf1* is much lower (if at all present) in the GP than in the surrounding tissues ([*Parker et al., 2007*] and our data), raising the question of whether the main local role of IGF1 is autocrine or paracrine. The fact that deletion of *Igf1* in the whole limb mesenchyme greatly diminishes chondrocyte hypertrophy by P7 (*Cooper et al., 2013*), whereas specific *Igf1* deletion in the chondrocyte lineage (including osteoblasts) affects bone mass accretion but not bone length by two weeks of age (*Govoni et al., 2007*), strongly suggests that the main source of IGF1 affecting the perinatal GP is the local soft tissues. Interestingly, one of the main downstream targets of insulin/IGF signaling, mechanistic target of rapamycin (mTOR), also plays a key role in chondrocyte hypertrophy, but it is not clear what are the sources and identities of the upstream molecules that activate this pathway in chondrocytes of the GP (*Chen and Long, 2014*; *Lai et al., 2013*; *Phornphutkul et al., 2008*; *Srinivas et al., 2009*).

Probably even more complex than growth regulation during normal development is growth regulation after an injury in a growing organ, as the deleterious effects of the injury have to be resolved while the rest of the body is growing. Tissue injury triggers activation of a specific group of genes characterized by a very quick response to signals such as cellular stress (*Bahrami and Drabløs, 2016*). Such genes are often referred to as immediate-early genes. Tissue injury often also triggers an inflammatory response characterized by immune cell recruitment and release of inflammatory cytokines, which have been shown to impair organ growth, including bones (*MacRae et al., 2006*; *Mårtensson et al., 2004*).

Elucidating the mechanisms by which local signals extrinsic to the GP modulate bone growth will be relevant to understanding regional differences in bone growth, both within and between species, and also for developing non-invasive therapies for congenital or acquired growth disorders. A major obstacle for exploring local bone growth regulation is a lack of animal models in which only cells outside the GP are manipulated. We show here that transient unilateral induction of cell death in the soft tissues of the postnatal limb impairs GP function and bone growth, leading to left-right limb length inequality. The growth defect is associated with reduced chondrocyte proliferation and hypertrophy, due to two paracrine signaling branches from the knee joint: (1) Injury-induced local inflammation reduces IGF1 in the knee fat pad, impairing mTOR and IHH signaling in the GP. Interestingly, this branch is active in neonates but not older mice with a secondary ossification center. (2) The injury response cascade is activated in the joint and prospective articular cartilage, affecting multiple signaling pathways in the adjacent resting zone (RZ) of the GP. Both IGF1 loss and the injury response pathway have a causative role in the reduction of bone growth, as combined maintenance of IGF1 expression and inhibition of the injury response rescues bone growth following cell death outside the neonatal GP, whereas either treatment alone does not. Our study reveals that neonatal bone growth is modulated by extrinsic signals from the fat pad and the articular cartilage, opening

new avenues for developmental and evolutionary studies, and for the correction of growth defects in humans.

## Results

### Growth of the left long bones is impaired following transient left-specific cell death in the adjacent mesenchyme around birth

As a means to study local bone grow control mechanisms, we developed a genetic model in mice in which transient cell death is induced specifically in the mesenchyme of the left limb, allowing the right limb to be an internal control (*Figure 1A*). A transgenic mouse line that drives Cre expression in the left lateral plate mesoderm under the control of the *Pitx2* asymmetric enhancer (*Shiratori et al., 2006*) was combined with an *R26*<sup>LSL-DTR</sup> strain that expresses diphtheria toxin receptor (DTR) following Cre-induced recombination (*Buch et al., 2005*) (Pit::DTR mice). As expected, Pit::DTR mice expressed DTR primarily in the left but not right limb mesenchyme, excluding muscles and blood vessels (*Figure 1A'*; *Figure 1—figure supplement 1A–B*). To model an acute injury at birth, a single systemic injection of DT was administered at postnatal day (P) 1 (P1-Pit::DTR model), resulting in cell death in the left hindlimb mesenchyme between 1 and 3 days post-injection (dpi), with little in the right limb (*Figure 1B* and not shown). Unexpectedly, however, cell death was not detected in the left GPs of the tibia and femur of P1-Pit::DTR mice, and the front of apoptotic

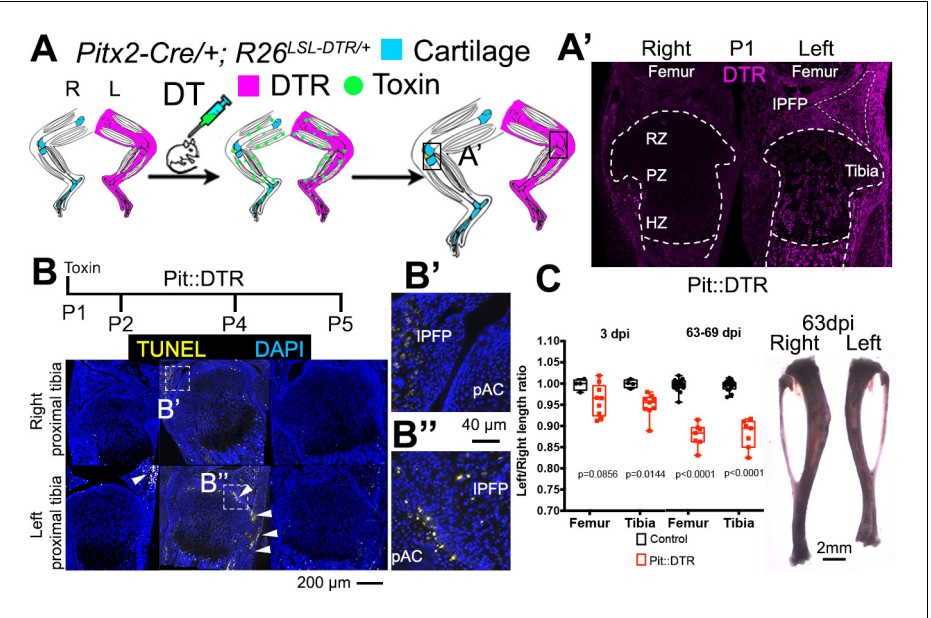

**Figure 1.** Postnatal growth of the left long bones is impaired following transient left-specific cell death in the adjacent mesenchyme. (**A**) Pit::DTR model for the induction of transient unilateral cell death in the limb mesenchyme. See *Figure 1—figure supplement 1* for a characterization of the *Pitx2-Cre* lineage. LSL= loxSTOPlox. (**A'**) Sagittal sections of Pit::DTR knees immunostained for DTR (approximate region indicated by the boxes in (**A**). RZ, PZ, HZ= resting, proliferative, hypertrophic zone. IPFP= infrapatellar fat pad. (**B–B''**) Time course of cell death induction (TUNEL, arrowheads) after DT injection at P1. Insets show details at the knee joint (pAC= prospective articular cartilage). The cytoplasmic signal in (**B'**) is unspecific background. (**C**) Skeletal preparation (P1-Pit::DTR tibiae) and plot of left/right length ratio for P1-Pit::DTR or control femur and tibia from mice collected as indicated. Analysis was done by 2-way ANOVA (alpha = 0.05, Bone Identity and Genotype as variables, p-value for Genotype was 0.0177 at 3dpi, < 0.0001 at 63-69dpi) followed by Sidak's posthoc multiple comparisons test (p-values shown in Figure).

The following figure supplement is available for figure 1:

**Figure supplement 1.** Characterization of the *Pitx2-Cre* derived cell lineage.

cells only reached the outermost layers of the prospective articular cartilage and perichondrium from P2 to P5 (*Figure 1B–B''*); the proportion of cells undergoing cell death in the perichondrial groove of Ranvier was 18 ± 7% at P2, n = 6). It is likely that DT (a ~60 kDa molecule) cannot diffuse efficiently into the postnatal GP, which is avascular (*Figure 1—figure supplement 1D*) and a very effective barrier to the diffusion of molecules larger than 10 kDa (*Williams et al., 2007*). To rule out an abnormal response of chondrocytes to DT, we cultured Pit::DTR tibiae from P1 mice in various concentrations of DT (5, 50 and 500 ng/ml) for 24 hr, to test whether a very high concentration of DT could overcome the potential diffusion barrier and ablate GP chondrocytes. We found that at the highest concentration of DT, cell death was triggered in some cells in the core of the GP (*Figure 1—figure supplement 1E*), confirming that postnatal chondrocytes can respond to DT.

Interestingly, despite sparing of the GP in the P1-Pit::DTR model, cell death in the surrounding tissues resulted in ~5% (0.3mm) length difference between the left and right femora and tibiae at three dpi that was not observed in control animals (either DT-injected *Pitx2-Cre; R26$^{+/+}$* or PBS-injected Pit::DTR mice), and that plateaued at ~11% (~2 mm) after 9 weeks (*Figure 1C*, n = 4 control and 9 experimental mice at 3dpi, n = 16 and 7 respectively at 63-69dpi). The forelimbs were not affected, as they express little DTR (*Figure 1—figure supplement 1C*).

The left specific reduction in bone growth took place on top of a transient systemic growth reduction with respect to control littermates, both in body weight and length of the right long bones (*Figure 2—figure supplement 1A–B*). The systemic growth reduction was likely caused by cell death in a region of the heart where *Pitx2-Cre* drives DTR expression (*Shiratori et al., 2006*) (*Figure 2—figure supplement 1C–C'*), as we observed that P1-Pit::DTR animals were lethargic for ~4 days after DT injection, including severe (sometimes lethal) hypophagia. As a consequence, the GPs of P1-Pit::DTR mice at P5 were smaller than in control animals (*Figure 2—figure supplement 1D–E'*). At the molecular level, one difference found between control and P1-Pit::DTR mice at P5 was a decrease in the synthesis of ECM in both the left and right GPs of P1-Pit::DTR mice, as *Agcn1* and *Col2a1* expression was downregulated compared to controls (*Figure 2—figure supplement 1F*). After 4dpi, however, experimental mice resumed normal feeding and started catching up with the controls. Importantly, by the end of the longitudinal growth period, the lengths of the right but not the left long bones had normalized in P1-Pit::DTR mice (*Figure 2—figure supplement 1B*), demonstrating the left-specific growth reduction is independent of the transient systemic effect. It is worth noting that unwanted recombination, as observed in the Pit::DTR heart, is a common caveat of most limb-targeting Cre lines, such as *Prrx1-Cre* and *Hoxb6-Cre,* which target regions of the head and gut (see JAX005584, 017981 and references therein). Unlike the bilateral approaches, however, our unilateral strategy allowed us to distinguish local and systemic effects by using the right limb as an internal control.

In conclusion, the specific growth defect in the left hindlimb bones of P1-Pit::DTR mice raised the interesting possibility that the injured tissues adjacent to the GP negatively modulate bone growth via a paracrine (i.e. extrinsic) mechanism.

## Reduced chondrocyte proliferation and hypertrophy underlie the left hindlimb growth defect in the P1-Pit::DTR model

As differential bone growth depends to a great extent on changes in chondrocyte proliferation and hypertrophy (*Wilsman et al., 1996*), we first tested whether these processes were altered in the left GP of P1-Pit::DTR animals. Indeed, the left/right ratio of EdU incorporation in the proliferative zone (PZ) of the proximal tibial GPs was reduced in the P1-Pit::DTR model compared to control animals at P3 (*Figure 2A*, n = 4 control and 7 experimental animals). Moreover, at P4-P5 we observed an obvious decrease in the height of the HZ but not the PZ (*Figure 2B*), as further revealed by in situ hybridization for the chondrocyte maturation marker *Col10a1* (*Figure 2C*, n = 3) and hematoxylin and eosin staining (*Figure 2D*, n = 4 control, 4 experimental). In addition, the formation and expansion of the secondary ossification centers in the left bones of P1-Pit::DTR mice at P4-5 was slightly delayed and progressed more slowly as compared to the right bones (*Figure 2E–F*, n = 5). Thus, not only proliferation and differentiation of chondrocytes, but also maturation of the whole skeletal unit, are impaired in the left hindlimb skeletal elements of P1-Pit::DTR animals as a consequence of transient cell death in the surrounding tissues.

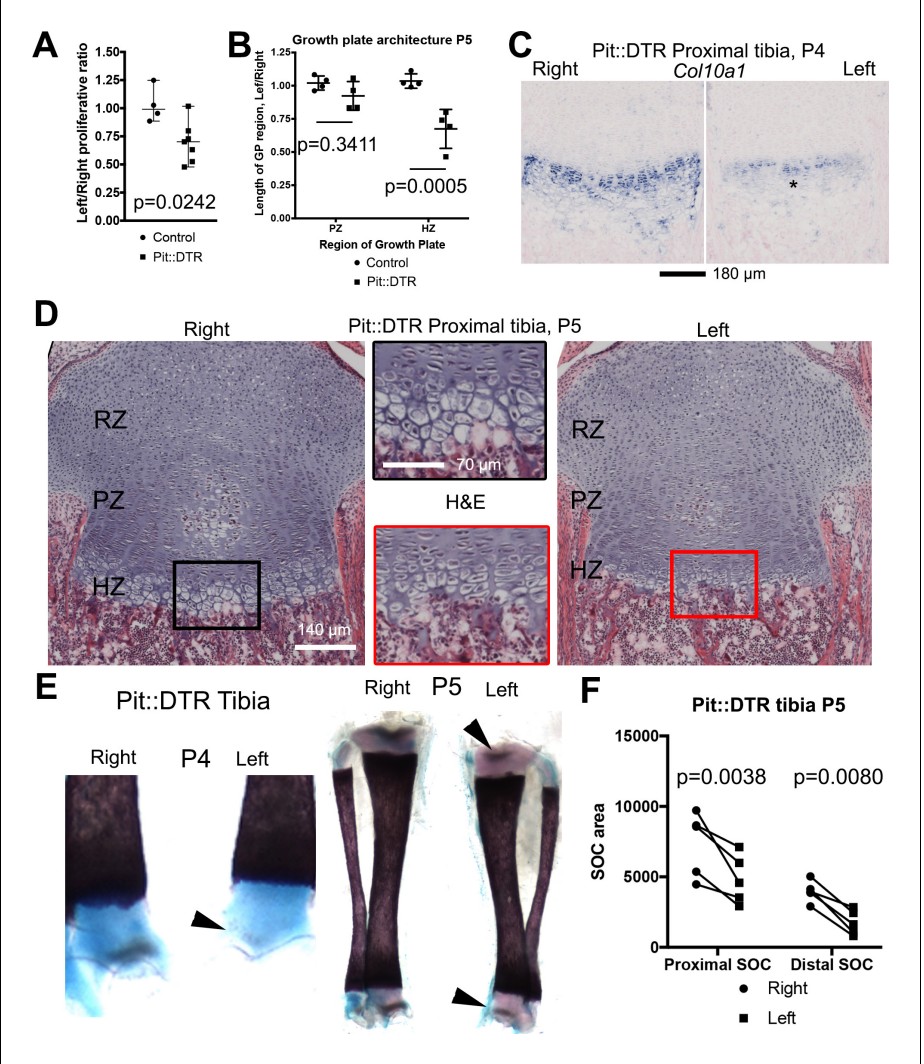

**Figure 2.** Reduced chondrocyte proliferation and maturation underlie the growth defect of the left bones of P1-Pit::DTR mice. (**A**) The fraction of EdU+ nuclei in the PZ was calculated for left and right proximal tibia GPs 2dpi and represented as L/R ratio (mean ± SD). p-value for unpaired two-tailed Mann-Whitney test between control and experimental ratios is shown. (**B, D**) The PZ and HZ of left and right GPs were measured on hematoxylin and eosin-stained sections of P1-Pit::DTR proximal tibia GPs, 4dpi (**D**), and represented as L/R ratios (**B**) 3–4 sections per GP, n = 4 mice, mean ± SD are shown). 2-way ANOVA (variables: Genotype and GP region, alpha = 0.05, p-value=0.0118 for Genotype) followed by Sidak's posthoc multiple comparisons test (p-values shown) was used. Insets in (**D**) show magnifications of the boxed regions. (**C**) RNA in situ hybridization for *Col10a1* in the proximal tibia GP at 3dpi. (**E, F**) The secondary ossification centers (SOCs, arrowheads in **E**) appear later and their subsequent area (quantified in **F**) is reduced in the left P1-Pit::DTR skeletal elements. Analysis was done by 2-way ANOVA (variables: SOC location and Side, alpha = 0.05, p-value=0.0003 for Side) followed by Sidak's posthoc multiple comparisons test (p-values shown in Figure). Asterisk= reduced expression. See also associated *Figure 2—figure supplement 1*.

The following figure supplement is available for figure 2:

**Figure supplement 1.** Bone growth impairment in Pit::DTR animals takes place on top of a systemic growth delay likely caused by injury-induced hypophagia.

## Multiple signaling changes related to chondrocyte proliferation and hypertrophy underlie the growth defect of the left bones

We first confirmed that changes in intrinsic growth regulation mechanisms were not the origin of the bone growth impairment in the P1-Pit::DTR model, by examining well-known signaling pathways. Importantly, HTCs did not display increased EGFR signaling (*Figure 3—figure supplement 1A*), which can be activated by DTR shedding (*Xu et al., 2004*) and impairs terminal chondrocyte differentiation and cartilage remodeling (*Zhang et al., 2011*; *Hall et al., 2013*). In addition, the potential ablation of osteoprogenitors in the perichondrium of P1-Pit::DTR animals was an unlikely cause of the reduced bone growth, as most of these cells were not TUNEL+ in our model (*Figure 3—figure supplement 1C–C'*). Moreover, a near-complete ablation of osteoprogenitors using an *Sp7* (Osx)-driven Cre (Osx::DTR model) failed to impair longitudinal bone growth, although it did reduce cortical thickness (*Figure 3—figure supplement 1D–F*). Finally, we did not detect increased cell death in the osteochondral junction that could explain the observed HZ shortening (*Figure 3—figure supplement 1G*).

To probe the link between soft-tissue signals and GP function, we next tested whether the signaling pathways known to affect chondrocyte proliferation and hypertrophy were altered in the left hindlimb GPs of P1-Pit::DTR mice. Since cell death receded by P5, we focused our analysis on the first days post-injection for the rest of the study. Significantly, we found that *Ihh* expression was reduced in the left compared to right GPs at P3, as was the expression of the HH target gene *Gli1* (*Figure 3A–A'*, n = 14/16 and 8/10, respectively). We did not, however, observe a change in expression of the IHH target *Pthlh* (*Figure 3—figure supplement 2A*). Given that the phosphatase SHP2 negatively modulates *Ihh* levels in cultured chondrocytes (*Guan et al., 2014*), we treated P1-Pit::DTR pups with an SHP2 inhibitor from P1 to P3. While the treatment rescued *Ihh* expression, it did not rescue bone growth (*Figure 3—figure supplement 2B–C'*), suggesting that other parallel signaling pathways were disturbed. Since mTOR complex 1 (mTORC1) is implicated in chondrocyte maturation and hypertrophy (*Lai et al., 2013*; *Phornphutkul et al., 2008*; *Srinivas et al., 2009*), we next performed immunohistochemical detection of p-S6, a readout of mTORC1 activity, and found that indeed p-S6 levels were reduced in the left PZ and pre-HZ from P3 until at least P5 (*Figures 3B* and *4*, n = 19/23, 7/8 and 3/3 at P3, P4 and P5, respectively). We further found that this downregulation could be mimicked in the right GP of P1-Pit::DTR mice and also in both hindlimbs of WT pups by treatment with the mTORC1 inhibitor rapamycin, which also stunts overall growth (*Figure 3—figure supplement 2D–E*). Since FGFs secreted by the perichondrium negatively regulate chondrocyte proliferation and early hypertrophy (*Liu et al., 2002*; *Karuppaiah et al., 2016*), we tested whether FGF signaling was altered, but did not detect an increase in FGF signaling in any region of the left GP of P1-Pit::DTR mice at P3 (*Figure 3—figure supplement 1B–B'*). On the contrary, *Fgfr3* expression was downregulated in the region of the left GP where the secondary ossification center forms, suggesting *Fgfr3* could be involved in the observed delayed formation and expansion of this structure. Moreover, decreased TGFβR2 signaling in the joint region has been shown to impair chondrocyte hypertrophy at embryonic stages via increased expression of the cytokine MCP5 (*Longobardi et al., 2012*), but we were not able to detect MCP5 mRNA or protein in our model at P3. On the other hand, RNA and protein levels of the WNT target *Lef1* were upregulated in the RZ of the left P1-Pit::DTR GP compared to right (*Figure 3C*; see also *Figure 4—figure supplement 2*, n = 5 at P3, 4 at P4, 3 at P5), which might impair PTHLH signaling and chondrocyte differentiation, as described (*Ray et al., 2015*; *Guo et al., 2009*). In summary, our results show that damage in the tissues adjacent to the early postnatal GPs results in alterations in multiple GP signaling pathways known to regulate bone growth.

## Inflammation reduces *Igf1* expression in the infrapatellar fat pad, leading to impaired mTORC1 and IHH signaling in the growth plate

We next undertook a candidate approach to explore the possible extrinsic influences that were responsible for the signaling changes in the experimental GPs. Since mTORC1 activity was impaired in P1-Pit::DTR mice and insulin/IGF signaling is the main bone-growth related pathway known to activate mTORC1 signaling, we examined the GH/IGF signaling axis. As the right limb serves as an internal control for the insulted left limb in our model, any left-specific defect could not be due to changes in a systemic factor(s) (be it GH or IGF1), unless there was a local change in the signal

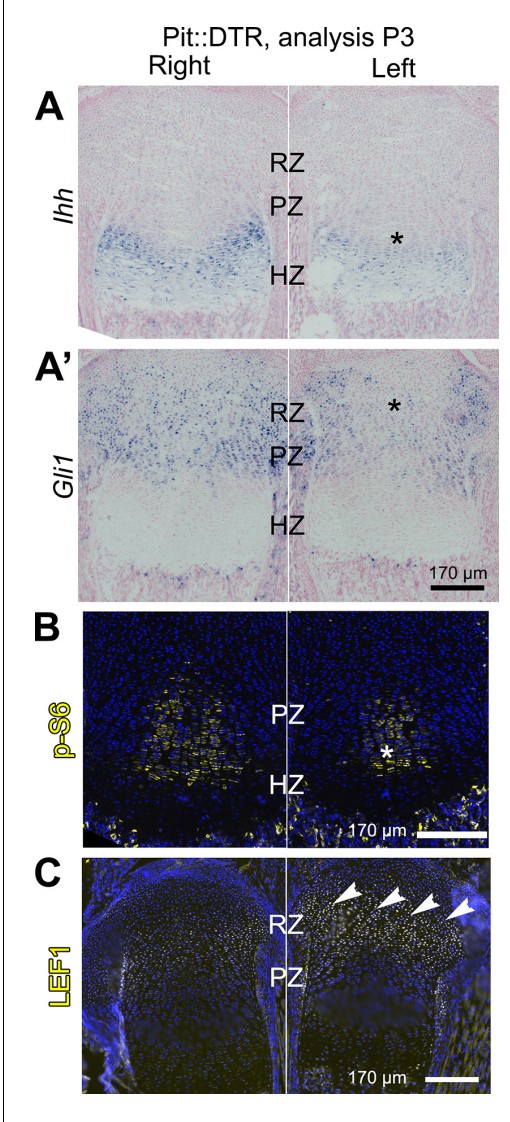

Pit::DTR, analysis P3

**Figure 3.** Multiple signaling pathways related to chondrocyte proliferation and maturation are altered in distinct regions of the left P1-Pit::DTR growth plate. (A–A') In situ hybridization for *Ihh* and the HH target *Gli1*, in the left and right proximal tibia of Pit::DTR mice, 2dpi. (B, C) Immunohistochemical staining for phosphorylated ribosomal protein S6 [(B), readout of mTORC1 activity] or for the canonical WNT target LEF1 (C), in the left and right proximal tibia of Pit::DTR mice, 2dpi. Note left-specific downregulation of p-S6 in the same region where *Ihh* is reduced. Arrowheads= ectopic expression, asterisks= reduced expression. See also associated *Figure 3—figure supplements 1* and *2*.

The following source data and figure supplements are available for figure 3:

**Source data 1.** Cortical bone parameters measured by μCT in P5 femora from Osx::DTR mice injected with PBS (Ctl) or DT (Exp) at P1.

*Figure 3 continued on next page*

transduction machinery that responds to those systemic factors. Furthermore, given that systemic GH/IGF1 does not play an important role in early postnatal skeletal growth (see Introduction), we decided to focus on local IGF signaling. Interestingly, we found that while *Igf1* was highly expressed in the right infrapatellar fat pad (IPFP), an extra-synovial adipose tissue located between the femur and the tibia, it was extremely and specifically downregulated in the left IPFP of P1-Pit::DTR mice at 2-3dpi (*Figure 4A*, n = 10/12), and recovered by 4dpi (*Figure 4A'*, n = 2). Furthermore, *Igf1* expression was not consistently altered in other tissues, such as the perichondrium, the muscle bundles or cells in the metaphyseal (shaft) region (*Figure 4—figure supplement 1A*). Similarly, we only detected minor to no changes in the expression of *Igf2* or the main IGF receptor gene *Igf1r* in the GP or surrounding tissues (*Figure 4—figure supplement 1B–C*), indicating that the IPFP was specifically affected by the insult. We further tested whether downstream signaling components were altered. In particular, since injury-induced inflammation (see below) could conceivably increase expression of suppressor of cytokine signaling 3 (SOCS3), which is known to exert a negative influence on IGF signaling (*Ahmed and Farquharson, 2010*), we performed SOCS3 immunohistochemistry on P1-Pit::DTR knee sections. However, we could not detect differential expression between the left and right GPs at P2 or P3 (*Figure 4—figure supplement 1D*), suggesting that it is the reduced availability of IGF1 from the IPFP that causes some of the signaling changes observed in the left GP. To test this possibility, we provided exogenous IGF1 to the left knee region by intraarticular injections in P1-Pit::DTR pups (P1-P3), and found that indeed mTORC1 signaling was consistently rescued in the PZ and pre-HZ of the GPs (*Figure 4B'* top, n = 5 P3-P5 pups). Curiously, *Ihh* and *Fgfr3* expression were also rescued in these animals (*Figure 4B'* bottom, n = 3 at P3, and *Figure 4—figure supplement 2A*), suggesting that both genes are downstream of IGF signaling.

To determine the cause of *Igf1* downregulation, we analyzed the cellular and molecular response to cell death in the left IPFP of P1-Pit::DTR mice. We observed left-specific neutrophil recruitment at P2-P4 that had mostly receded by P5, thus precisely correlating with transient *Igf1* downregulation (*Figure 4C–C'* n = 5, 5, 4, 5 at P2, P3, P4, P5). Moreover, macrophages, already very abundant in the IPFP in resting conditions, were additionally recruited to the left IPFP

*Figure 3 continued*

**Figure supplement 1.** Bone growth impairment in P1-Pit::DTR animals is not caused by increased FGF or EGF signaling in the HZ, nor by ablation of osteoprogenitors in the perichondrium or increased cell death in the osteochondral junction.

**Figure supplement 2.** Characterization of the role of IHH and mTOR in the signaling and growth defects of P1-Pit::DTR mice.

(*Figure 4—figure supplement 3A–A'*). As neutrophils are an important feature of joint inflammatory diseases (*Wright et al., 2010*; *Wipke and Allen, 2001*), we tested whether neutrophils were responsible for *Igf1* downregulation by blocking their recruitment with the NIMP-R14 neutralizing antibody (*Lopez et al., 1984*) during P1-P3 in P1-Pit::DTR mice. Neutrophil blockade was effective for at least 14 hr (*Figure 4D*, n = 3), and significantly correlated with partial recovery of *Igf1* expression in the IPFP (*Figure 4D'*, n = 4), without inhibiting cell death in this region (*Figure 4—figure supplement 3B*). The partial recovery of *Igf1* after neutrophil inhibition indicates that additional cell types inhibit its expression. Importantly, the recovery of *Igf1* expression correlated with increased levels of p-S6 and *Ihh* in the pre-HZ (*Figure 4D''*, n = 3, and not shown), demonstrating that proper signaling in the pre-HZ requires IGF1 production from the IPFP. To further probe the role of immune-cell recruitment in *Igf1* downregulation, we injected lipopolysaccharide (LPS) into the joint of WT animals, which caused a strong (but patchy) local recruitment of neutrophils to the IPFP. Of significance, this recruitment led to the predicted downregulation of *Igf1* in the areas with highest accumulation of neutrophils (*Figure 4—figure supplement 3C*). This effect took place in the absence of cell death (*Figure 4—figure supplement 3D*), suggesting that immune cells (probably neutrophils) secrete factors that reduce *Igf1* expression. Notably, we found that TNFα-expressing cells (some of them neutrophils) were present in the left and not the right IPFP in P1-Pit::DTR mice at P2, and that neutrophils accumulated TNFα in what seemed to be the Golgi apparatus (*Figure 4C'''*). TNFα has been shown to reduce *Igf1* expression in vascular smooth muscle (*Anwar et al., 2002*), raising the possibility that this cytokine is in part responsible for the observed *Igf1* downregulation in the left IPFP of P1-Pit::DTR mice. Interestingly, partial restoration of *Igf1* expression by neutrophil blockade did not rescue *Fgfr3* expression (*Figure 4—figure supplement 2B*), suggesting that full restoration of *Igf1* is necessary for normal *Fgfr3* expression. Despite IGF1 supplementation rescuing several critical pathways, neither IGF1 injection nor neutrophil blockade were sufficient to rescue the decreased left bone growth at P4 (*Figure 4E–F*, n = 5 experimental and 5 control animals for each treatment), indicating that an IGF1-independent pathway also contributes to the growth defect. Indeed, expression of *Lef1* in the left GP remained upregulated after IGF1 injection or neutrophil blockade (*Figure 4—figure supplement 2A–B*), raising the possibility that increased LEF1 activity contributes to the growth defect.

## The injury response is activated in the articular cartilage

Given the importance of the immediate-early damage response upon injury, we tested its activation and indeed detected it in the left joint region of P1-Pit::DTR mice, not only in the IPFP but also in the prospective articular cartilage (pAC). Specifically, we observed that expression of *Early growth response gene 1* (*Egr1*) was increased in the left IPFP and pAC of P1-Pit::DTR mice at P2 (1dpi) (*Figure 5A*, n = 2). *Egr1* expression in the left IPFP receded by P3, but remained highly upregulated in the left pAC through P4, receding by P5 (*Figure 5B–C*, *Figure 5—figure supplement 1A–B*, n = 3, 2, 2, respectively). Interestingly, p-S6 expression, while downregulated in the pre-HZ (*Figures 3* and *4*), was upregulated in the left pAC at P3 and P4 and receded by P5 (*Figure 5B–C*, n = 33, 14 and *Figure 5—figure supplement 1B'*, n = 2), suggesting transiently increased mTORC1 signaling in this region, as observed in osteoarthritis (*Pal et al., 2015*). Other injury response markers followed a dynamic similar to *Egr1* in the left pAC of P1-Pit::DTR mice. For example, we detected ectopic expression of *Hif2a* (*Figure 5—figure supplement 1A*), which has a prominent role in cartilage destruction during osteoarthritis and rheumatoid arthritis (*Ryu et al., 2014*; *Husa et al., 2010*). In addition, *Il6*, one of the main mediators of *Hif2a*-dependent cartilage destruction (*Ryu et al., 2014*), was upregulated in the left pAC (*Figure 5—figure supplement 1A*), and was the likely cause of the ectopic JAK1/2-dependent expression of p-STAT3 we detected in the pAC at P3 and P4 (*Figure 5B–C*, n = 13 and 3; *Figure 5—figure supplement 1C*), and that receded by P5 (not shown).

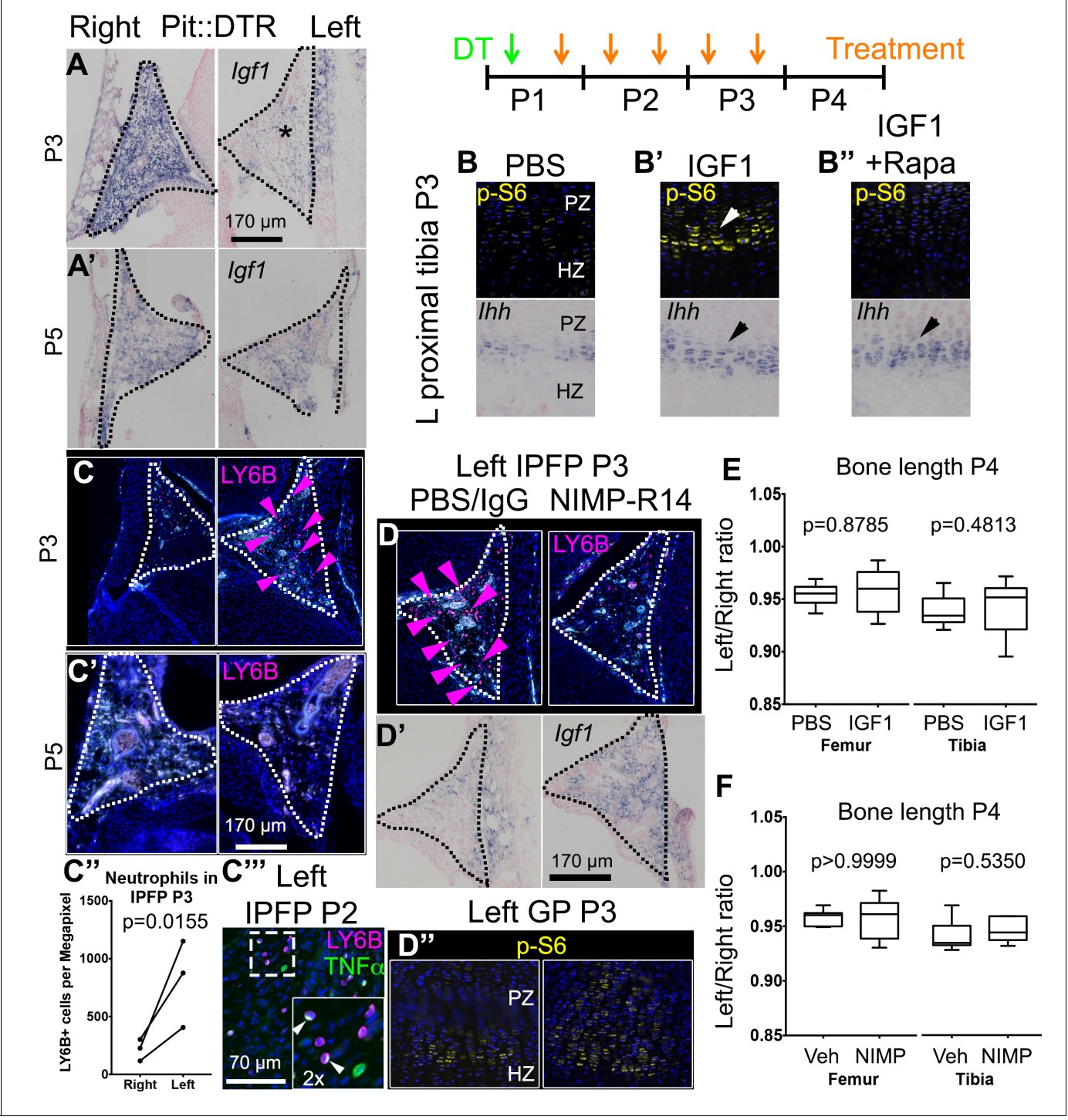

**Figure 4.** Injury-induced inflammation in the left infrapatellar fat pad reduces local *Igf1* expression in the DT-impaired growth plate. (**A, A'**) In situ hybridization for *Igf1* in the infrapatellar fat pad (IPFP, dotted lines) of left and right P1-Pit::DTR knees, at P3 (**A**) and P5 (**A'**). (**B–B''**) Immunohistochemistry for the mTORC1 readout p-S6 (top), and RNA in situ hybridization for *Ihh* (bottom) in left P1-Pit::DTR GPs at P3, following intraarticular injection of PBS (**B**), IGF1 (**B'**) or IGF1 combined with i.p. injection of the mTORC1 inhibitor rapamycin (**B''**). (**C–C'''**) Immunohistochemistry for the neutrophil marker LY6B in the IPFP (dotted lines) of left and right P1-Pit::DTR knees, at P3 (**C**), P5 (**C'**) and P2 (**C'''**). The inset in (**C'''**) is a 2x magnification showing that some neutrophils express TNFα in a cellular compartment. Turquoise signal= autofluorescent cells. The graph in (**C''**) represents the density of LY6B+ cells in left and right P1-Pit::DTR IPFP at P3 (n = 3 mice, 3–4 sections per animal). A ratio paired t-test was used to offset the variability between absolute measurements. (**D–D''**) Immunoblockade of neutrophil infiltration with NIMP-R14 antibody after DT injection (**D**) rescues *Igf1* expression in the left IPFP (**D'**), as well as mTORC1 signaling in the GP, 2dpi (**D''**). (**E, F**) Quantification of bone length at P4, expressed as

*Figure 4 continued on next page*

*Figure 4 continued*

Left/Right ratio, for vehicle (Veh, either PBS or IgG)-treated and IGF1- (E) or NIMP-R14-treated (F) mice (unpaired two-tailed Mann-Whitney test). See also associated *Figure 4—figure supplement 1–3*.

The following figure supplements are available for figure 4:

**Figure supplement 1.** Characterization of the IGF signaling axis in P1-Pit::DTR mice.
**Figure supplement 2.** IGF1 supplementation or neutrophil immunoblockade do not rescue all the signaling changes in the left RZ of P1-Pit::DTR mice.
**Figure supplement 3.** Multiple immune cells are recruited to the infrapatellar fat pad of Pit::DTR mice, which correlates with local *Igf1* downregulation.

Importantly, expression of *Egr1,* p-S6 and p-STAT3 was not due to inflammation in the IPFP, as neutrophil blockade or injection of IGF1 did not preclude activation of the injury response in the pAC of P1-Pit::DTR mice (*Figure 5—figure supplement 1C* and not shown). Associated with the injury response, histological analysis revealed extensive and persistent cell loss in the pAC/AC from P4 to at least P27 (*Figure 5D–E*, n = 3 and 4). Of note, most signaling changes extended beyond the area of cell death (i.e. the pAC), almost reaching into the RZ of the GP (compare TUNEL and p-STAT3 in *Figure 5B–C*). Since cells in the pAC/AC do not normally give rise to GP chondrocytes (*Kozhemyakina et al., 2015*), this set of results raised the possibility that the injury response in the pAC triggers a paracrine-signaling cascade that contributes to the signaling defects observed in the RZ.

## Combined inhibition of the injury response cascade and intraarticular IGF1 injection rescues femur growth

To test the role of the injury response in the altered GP signaling in P1-Pit::DTR mice, we performed pharmacological rescue experiments. Significantly, in vivo treatment with the JAK1/2 inhibitor ruxolitinib (*Fridman et al., 2010*) precluded p-STAT3 and LEF1 upregulation in the articular region of P1-Pit::DTR mice (*Figure 6—figure supplement 1A–A'*). However, like IGF1 injection, ruxolitinib failed to rescue bone growth (*Figure 6A–B*, n = 5), perhaps because mTORC1 signaling remained downregulated in the pre-HZ (*Figure 6—figure supplement 1C'*). Finally, we tested whether redundancies exist between the fat pad-HZ axis and the pAC-RZ axis by administering ruxolitinib and intraarticular IGF1 either individually or in combination in the same litters. Notably, unlike single treatments, the combined treatment had a significant rescue effect (*Figure 6A–B*), although only some femora (3 out of 11) and none of the tibiae showed complete rescue (see Discussion).

## Left hindlimb bone growth is impaired following transient left-specific cell death in the adjacent mesenchyme after the secondary ossification center forms

Given that during the first two postnatal weeks the local extrinsic environment of the murine GP changes from having one to two adjacent ossification centers (*Pannier et al., 2010*), and therefore is located farther from the IPFP and potentially more exposed to diffusing molecules from the vasculature, we tested the effect of DT injection at P14 in Pit::DTR mice. Similar to the P1 injections, analysis of cell death 3-4dpi in P14-Pit::DTR mice revealed apoptotic cells only in the surrounding soft tissues, and not the GP (*Figure 7A*, n = 4). As in the P1-Pit::DTR model, the height of the left HZ zone at 4dpi was reduced in P14-Pit::DTR mice compared to controls (*Figure 7B*, n = 2), and the hindlimbs developed a progressive left-right asymmetry over time, although it was non-significant at P20 (~7–13% by P57, *Figure 7F–G*, n = 3–4 per genotype at P20, n = 7 per genotype at P57). Furthermore, although neutrophils infiltrated the knee joint, mTORC1 activity and *Ihh* signaling were not reduced at 3-4dpi in the left HZ as compared to the right in P14-Pit::DTR mice, unlike in the P1-Pit::DTR experiment (*Figure 7C*, n = 4 and not shown). Interestingly, we found that *Igf1* is no longer expressed in the right nor the left IPFP at P17-18 (*Figure 7D*, inset, n = 3), suggesting that mTORC1 activity in the HZ eventually becomes independent of this extrinsic source of IGF1. On the other hand, similar to P1-Pit::DTR mice, *Egr1* and LEF1 were ectopically activated in the left cartilage of

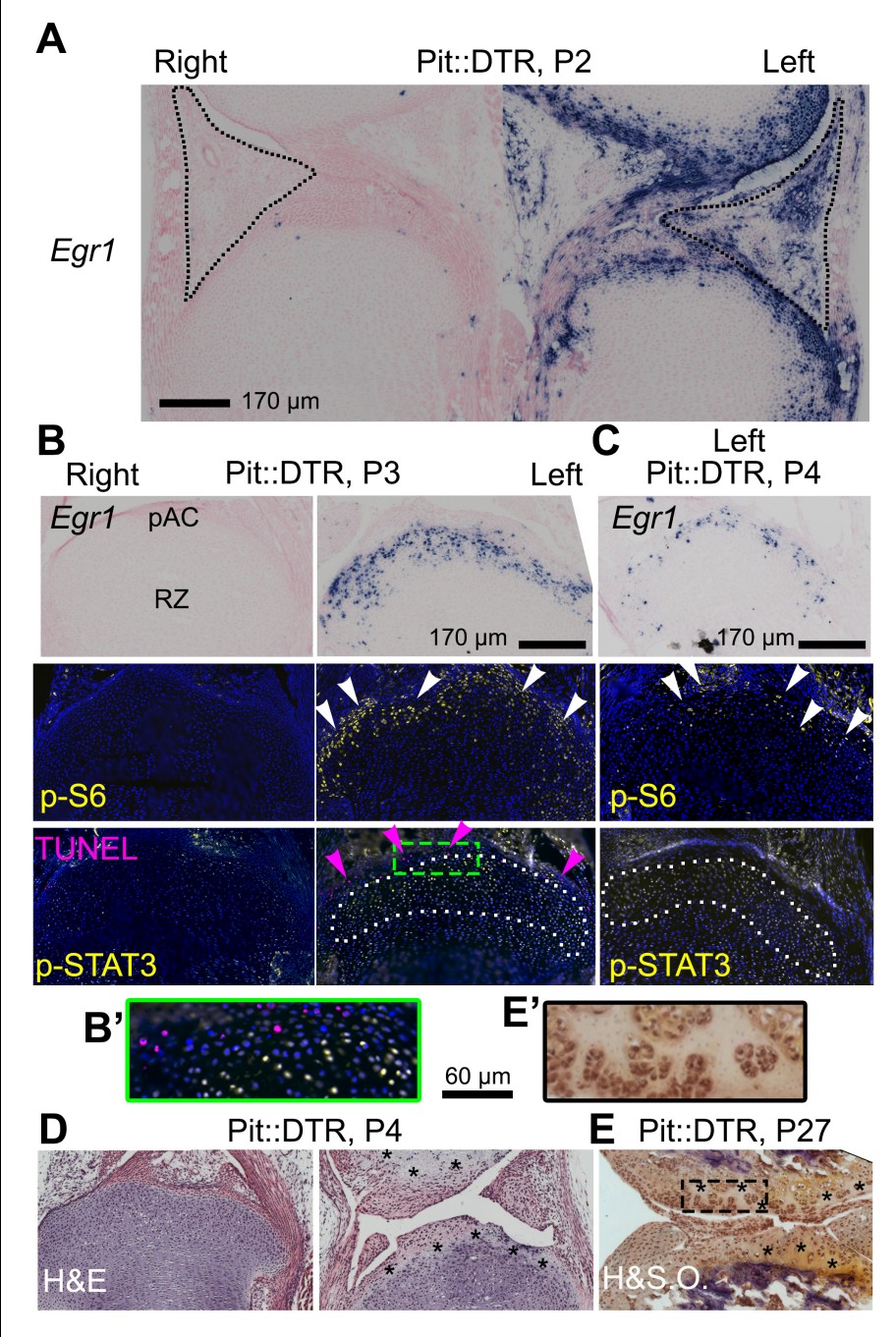

**Figure 5.** The injury response in the articular cartilage of P1-Pit::DTR mice correlates with signaling changes in the resting zone and impaired bone growth. (A) Expression of the immediate-early marker *Egr1* 1dpi. Dotted area= IPFP. (B, C) Expression of *Egr1* and the indicated signaling effectors (white arrows) and cell death (pink arrows) in the prospective articular cartilage (pAC), two (B) and three dpi (C). Dotted area= extent of p-STAT3 signal. (D, E) Hematoxylin and eosin (H&E) and hematoxylin and safranin O (H&SO) analysis of the pAC/AC, showing extensive damage at P4 (D), persistent at P27 (E). Magnifications of the boxes in (B) and (E) are shown in (B') and (E'), respectively. See also associated *Figure 5—figure supplement 1*.
The following figure supplement is available for figure 5:

**Figure supplement 1.** Characterization of the injury response in P1-Pit::DTR mice.

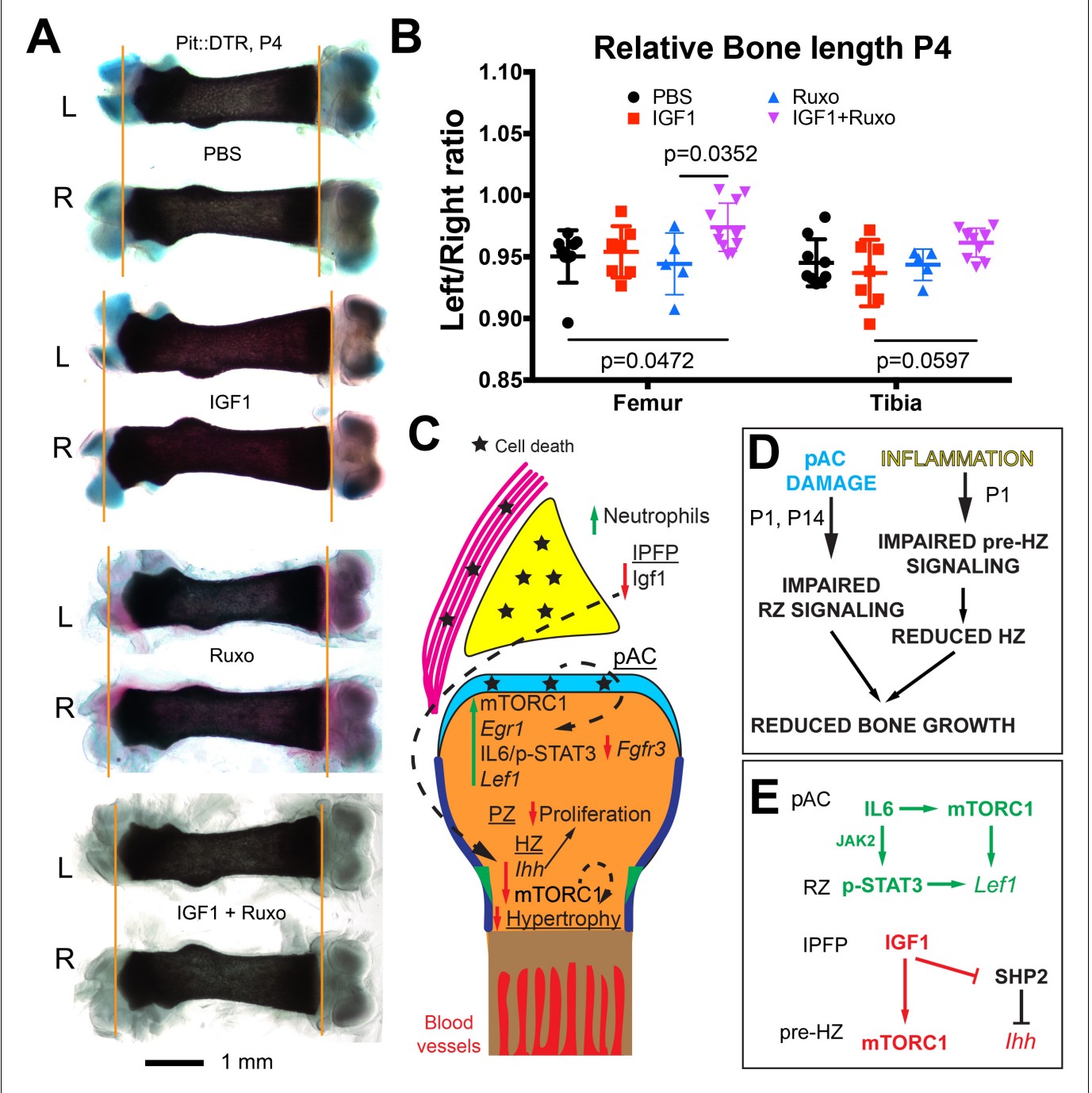

**Figure 6.** Combined inhibition of the injury response cascade and intraarticular IGF1 injection rescues DT-impaired femur growth. (A, B) Representative femur preparations (A) and quantification of left/right length ratio of P1-Pit::DTR bones at P4 (B) after in vivo treatment with the indicated substances (n = 8, 7, 5, 11 for PBS, IGF1, Ruxo, IGF1+Ruxo, respectively). The parallel treatments were compared by two-way ANOVA with Bone and Treatment as variables (alpha = 0.05, p=0.0066 for Treatment, 0.0678 for Bone), followed by Sidak's posthoc multiple comparisons test (only the p-values lower or close to 0.05 are shown in the Figure). (C, D) Summary of signaling changes in the IPFP and pAC after mesenchymal cell death outside the GP, and their interaction with GP signaling and bone growth. In (D), the postnatal stages at which each pathway operates are indicated. (E) Speculative model for the regulation of *Lef1* and *Ihh* in the experimental GP. Green/Red lettering indicates, respectively, pathways up/downregulated following inflammation and injury response. See also associated *Figure 6—figure supplement 1*.

The following figure supplement is available for figure 6:

**Figure supplement 1.** Inhibition of JAK1/2 or mTORC1 activity prevents the upregulation of LEF1 in P1-Pit::DTR mice.

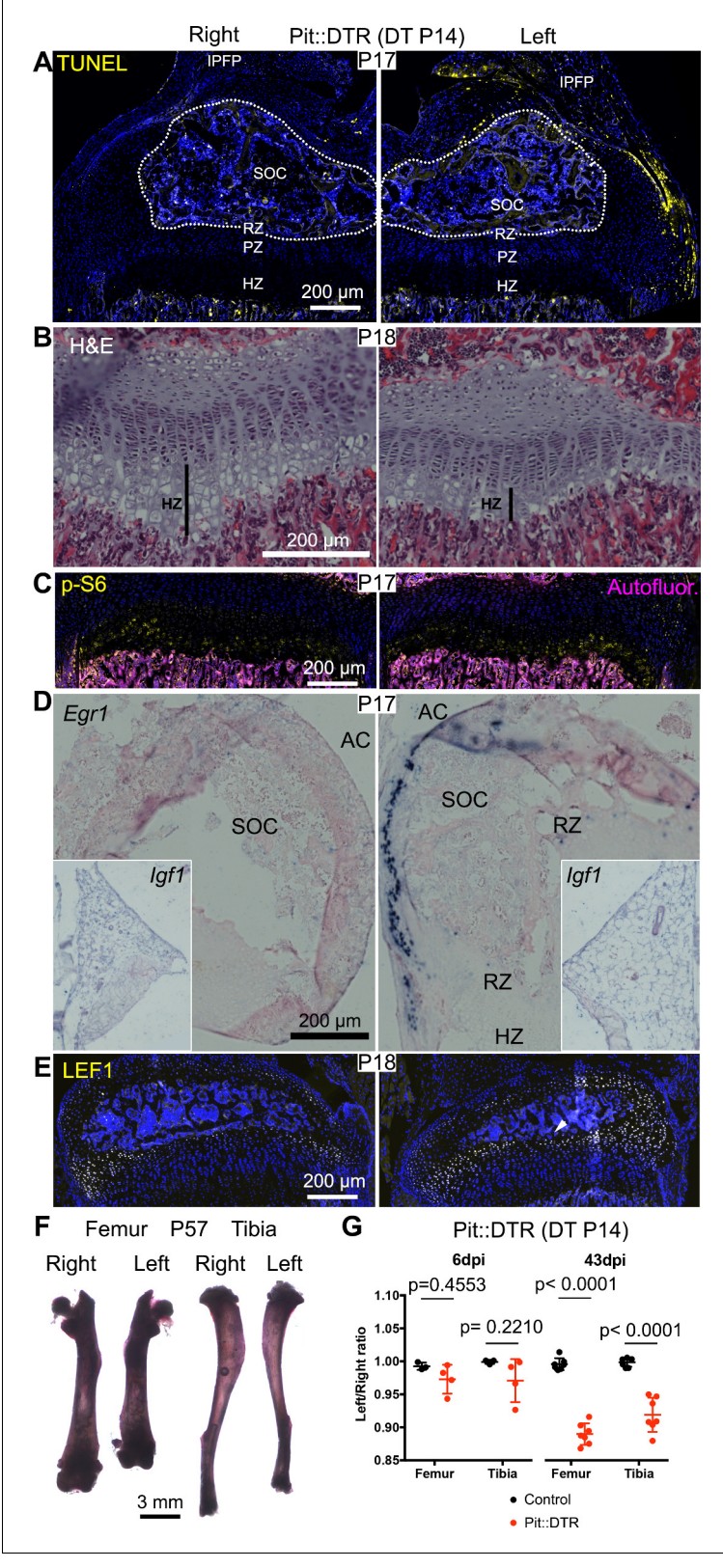

**Figure 7.** Induction of soft-tissue cell death at P14 also impairs bone growth. (**A**) TUNEL staining (arrowheads) in sagittal section of P17 proximal tibia from Pit::DTR mice injected DT at P14 (P14-Pit::DTR). SOC= secondary ossification center (outlined). (**B**) Hematoxylin and eosin-stained sagittal sections of P18 distal femur from P14-Pit:: DTR mice. Vertical bars indicate the length of the HZ. (**C**) Immunostaining for p-S6 in the GP of P14-Pit::DTR mice

*Figure 7 continued on next page*

*Figure 7 continued*

at P17. (D, E) In situ hybridization for *Egr1* (D) and *Igf1* (D, insets) and LEF1 immunostaining (E) in knees from P14-Pit::DTR mice. (F) Alizarin red staining of the skeletal elements from a representative P14-Pit::DTR mouse at P57. (G) Quantification of the left/right length ratio for femora and tibiae of P20 and P57 Control and P14-Pit::DTR mice. Analysis was done by 2-way ANOVA (alpha = 0.05, Bone Identity and Genotype as variables, p-value for Genotype was 0.1970 at P20 and < 0.0001 at P57) followed by Sidak's posthoc multiple comparisons test (p-values shown in Figure). Arrowheads denote ectopic expression. AC= articular cartilage. RZ, PZ, HZ= resting, proliferative, hypertrophic zones. IPFP= infrapatellar fat pad. See also associated *Figure 7—figure supplement 1*.

The following figure supplement is available for figure 7:

**Figure supplement 1.** Developmental description of infrapatellar fat pad formation and correlation with limb allometry.

---

P14-Pit::DTR mice (*Figure 7D–E*, n = 4). In conclusion, we observed a long-term impact on bone growth of a mesenchymal injury at P1 and P14, although the short-term mechanisms were only partially shared between both stages.

## Temporal sequence of infrapatellar fat pad development and its correlation with limb allometry

Given the new trophic role that we ascribe to the IPFP, the temporal differences we observed in *Igf1* expression in this tissue prompted us to characterize the appearance, cellular composition and expression of *Igf1* in the IPFP during mouse hindlimb development (*Figure 7—figure supplement 1A*). The IPFP appeared as a defined region at around E17.5, and this coincided with the accumulation of a high level of *Igf1* expression in the area as compared to the surrounding mesenchymal tissues. While perivascular stem cells (CD44/CD90+) (*Hindle et al., 2017*) were detected in the interzone area even before the IPFP was formed, adipocytes (detected either by hematoxylin and eosin staining or expression of FABP4) were not clearly seen until P3-P4, indicating that adipocytes are not the main source of *Igf1*. As expected, *Igf1* expression levels gradually decreased postnatally until they were barely detectable at P8 and not detected at P12. Intrigued by the dynamic expression of *Igf1* in the IPFP, we finally explored whether the dynamics correlate with any change in the ratio of the lengths of the radius (far from any fat pad, see *Figure 7—figure supplement 1A'*) and the tibia during development. Interestingly, we observed that the radius grows more slowly than the tibia at the stages when *Igf1* levels are high in the IPFP, whereas the ratio stabilizes after P8, roughly coinciding with the cessation of *Igf1* expression (*Figure 7—figure supplement 1B*). This correlation is consistent with the hypothesis that paracrine signals from the IPFP play a role in the establishment of body proportions, a possibility with clinical and evolutionary implications (see Discussion).

## Discussion

Our study demonstrates that transient cell death in the early postnatal left hindlimb mesenchyme surrounding the GPs leads to reduced long bone growth due to altered signaling from two tissues in the adjacent knee joint. Only a few recent studies have addressed the regulation of longitudinal bone growth by the surrounding tissues, and showed that TGFβR2 signaling from the joint interzone stimulates chondrocyte hypertrophy in the GP (*Longobardi et al., 2012*), and FGFs produced by the perichondrium negatively modulate bone growth in a GP non-autonomous fashion (*Liu et al., 2002*; *Karuppaiah et al., 2016*). Of significance, our study identifies two neonatal joint components that can modulate bone growth, at least following injury: the IPFP and the pAC/AC. Taken together, our data support a model whereby extrinsic signals from the IPFP and/or pAC/AC modulate distinct GP-intrinsic signaling pathways, leading to reduced bone growth, even in the absence of intrinsic damage to the GP (*Figure 6C–D*). While we think this model can be generally applied, the apparently milder rescue of the tibia compared to the femur (*Figure 6B*) could mean that each bone's response to the same growth-modulating cues varies with the developmental stage in a different manner. Indeed, the specific effect of limb immobilization on the growth of different bones has recently been shown to depend on the time of treatment in chicken and crocodile embryos (*Pollard et al., 2017*).

By inducing cell death outside the GP, we found a prolonged long bone growth defect in vivo that was associated with inflammation in the knee joint. This result is reminiscent of in vitro studies showing that pro-inflammatory cytokines can exert a negative influence on bone growth that continues well beyond the cytokine exposure period (*MacRae et al., 2006*). As in our in vivo study, the inflammation-dependent in vitro growth defect was only partially rescued by IGF1 treatment (*Mårtensson et al., 2004*), suggesting that inflammation impairs bone growth via both IGF-dependent and -independent pathways.

Using our P1-Pit::DTR mice, we also uncovered that loss of mTORC1 signaling in the pre-HZ correlates with loss of *Ihh* expression, and that restoration of IGF1 levels in the joint rescues both pathways. At the same time, SHP2 inhibition in P1-Pit::DTR mice rescues *Ihh* expression but not mTORC1 activity, and mTORC1 inhibition does not reduce *Ihh* expression in WT animals (*Sanchez et al., 2009*), nor does it impair IGF1-mediated rescue of *Ihh* expression in Pit::DTR mice (*Figure 4B''*). Therefore, we speculate that IGF1 activates *Ihh* expression via an mTORC1-independent pathway, possibly by inhibition of SHP2 (*Figure 6E*). Notably, the requirement of IGF signaling for *Ihh* expression seems to be restricted to a narrow perinatal window, as *Igf1r* deletion in the cartilage only reduces *Ihh* expression when triggered at P5 but not prenatally (*Wang et al., 2011*), and we show that *Igf1* is no longer expressed in the IPFP by P12.

Unlike in the pre-HZ, we found mTORC1 activity was increased in the left pAC and RZ of P1-Pit:: DTR mice. Since mTORC1 was induced despite reduction of local IGF1 levels, it follows that another signal must be responsible for mTORC1 activation in the AC/RZ. One candidate is IL6 expressed ectopically in the AC, as this cytokine can activate mTORC1 in some cell types (*Kim et al., 2008*). Interestingly, inhibition of mTORC1 activity dampened LEF1 levels in the RZ without impairing STAT3 phosphorylation, (*Figure 6—figure supplement 1A'', C''*), while JAK1/2 inhibition also precluded LEF1 upregulation, without impeding p-S6 activation (*Figure 6—figure supplement 1A',C'*). These results suggest that both mTORC1 and JAK/STAT signaling pathways are independently required to activate *Lef1* expression (*Figure 6E*).

We speculate that the local influence exerted by the joint region could at least in part explain poorly understood phenomena observed in the field of bone growth, such as the fact that the proximal tibia and distal femur GPs (i.e. closest to the fat pad) grow faster than the distal tibia or proximal femur (*Digby, 1916*; *Payton, 1932*; *Moss-Salentijn, 1974*). Our proposal is biologically relevant because differential GP growth has been recently suggested to be evolutionary optimized to achieve energy-efficient scaling of the growing bones by minimizing the remodeling of ossified cortical structures (*Stern et al., 2015*). While intrinsic differences in the GP likely exist and account in part for differential growth of proximal and distal GPs, classic experiments showed a change in growth rate upon proximo-distal transposition of the GPs within a bone (*Moss-Salentijn, 1974*; *Hert, 1964*), strongly suggesting the involvement of local extrinsic factors. Fat pads are probably one of the main participants in this process, as suggested by our study and those of others showing that adipocyte-secreted signals can stimulate long bone growth in the absence of growth hormone (*Shtaif et al., 2015*). In this regard, our finding that *Igf1* is almost absent from the IPFP by P8 correlates with the fact that the femur and tibia grow faster than the rest of the body (including forelimb bones) until roughly the same stage (*Figure 7—figure supplement 1B* and [*Roselló-Díez and Joyner, 2015*]). The GP likely becomes independent from the IPFP after ~P8 (due both to the intervening presence of the secondary ossification center as well as to cessation of *Igf1* expression in the fat pad), which could also explain why it takes longer for bone growth to be significantly impaired when the injury is induced at P14 *vs.* P1. Finally, we cannot exclude the possibility that the IPFP and/or other local tissues hosting cells with osteochondrogenic potential, such as the synovium, bone marrow or the perichondrial groove of Ranvier (*Hindle et al., 2017*; *Yang et al., 2013*; *Karlsson et al., 2009*; *Chung and Xian, 2014*), can contribute to bone growth not only with paracrine signals, but also with progenitor cells (see for example Fig. S9 in [*Yang et al., 2013*]), such that cell death within these tissues in Pit::DTR mice depletes a pool of cells that participates in recovery of the injured bones.

In conclusion, further studies using mouse models should confirm and expand the repertoire of local extrinsic regulators of bone growth. This repertoire would be a valuable resource for evolutionary studies addressing the change of body proportion across phyla, and it could potentially be harnessed to develop improved therapies to correct local long bone growth defects.

## Materials and methods

### Mouse strains

The *Pitx2-Cre* (*Shiratori et al., 2006*) (RRID:IMSR_RBRC03487, kind gift of Dr. H. Hamada), *Sp7-tTA, tetO-EGFP/Cre* (*Rodda and McMahon, 2006*) (RRID:IMSR_JAX:006361), *Pthlh^{lacZ}* (*Chen et al., 2006*) (RRID:MGI:5519222, provided by Dr. Chitra Dahia) and *R26^{LSL-DTR}* (*Buch et al., 2005*) (RRID: IMSR_JAX:007900) mouse lines were maintained in an outbred Swiss Webster background and genotyped as previously described. Noon of the day of vaginal plug detection was considered E0.5. The equivalent of E19.5 is referred to as P0. All animal studies were performed under an approved Institutional Animal Care and Use Committee mouse protocol according to MSKCC institutional guidelines.

### Diphtheria toxin injection

A stock solution of DT (Sigma) was prepared at 40 ng/μl in sterile PBS, aliquoted and stored at −80°C until used. A working solution (5 ng/μl in PBS for P1 and 15 ng/μl for P14) was prepared and 15 ng/g injected subcutaneously (scruff of the neck), using a 25 μl syringe (Hamilton).

Note: For all rescue experiments, it was important to determine that any putative rescue was not actually due to defective DT injection. Therefore, only the specimens that showed obvious signs of mesenchymal death (e.g. a characteristic deformity of the left foot and/or AC damage on histological examination) were used to assess the effect of drugs on the P1-Pit::DTR phenotype.

### Skeletal preparations and measurements

Staining of cartilage and bone was performed as described (*Rigueur and Lyons, 2014*). For young mouse pups (≤P5), ossified bone length was measured on digital microphotographs using the line tool in Adobe Photoshop. For adolescent and adult mice, the limbs were dissected out, skinned and incubated in 2% KOH to remove the soft tissues. Individual bones were then measured using digital calipers (EZCal from iGaging). Tibiae were measured from the intercondylar eminence to the distal articular surface, while femora were measured from the trochanteric fossa to the intercondylar fossa.

### Micro-CT

Femora from Osx::DTR mice, injected either with PBS or DT at P1, were collected at P5 and processed for skeletal staining as described above. After photographing, they were progressively transferred to 70% EtOH and stored at 4°C until scanning. Before scanning, the samples were allowed to reach room temperature. A 0.6 mm region of the mid-diaphysis (right below minor trochanter) was scanned on a Scanco μCT 35 system (Scanco Medical, Brüttisellen, Switzerland) using a 3.5 μm voxel size, 55KVp, 0.36 degrees rotation step (180 degrees angular range) and a 400 ms exposure time per view (performed in 70% EtOH). The Scanco μCT software (HP, DECwindows Motif 1.6) was used for 3D reconstruction and viewing of images. After 3D reconstruction, volumes were segmented using a global threshold of 0.3 g/cc. Tissue mineral density (TMD), cortical area fraction (Ct.Ar./Tt. Ar.), and thickness of the cortex (Ct.Th.) were calculated for the cortical bone.

### Drugs and growth factors

- The SHP2 inhibitor (NSC-87877, Tocris) was dissolved in sterile PBS at ~4.5 mM (2 mg/ml). Daily i.p. injections (5–10 mg/kg) were performed using a 25 μl syringe (Hamilton).
- Rapamycin (Cayman) stock was dissolved in EtOH at 50 mg/ml. The working solution was prepared at 0.5 mg/ml in PBS, and mice were injected with 1.5 μg/g s.c. once per day, using a 25 μl syringe (Hamilton).
- Ruxolitinib (Selleck) was solubilized in DMSO at 60 mg/ml, and then diluted to 6 mg/ml in 0.5% hydroxypropylmethylcellulose (Sigma) for oral gavage (60 mg/kg, daily) (*Butchbach et al., 2007*).
- rmIGF1 (R and D) was reconstituted to 250 ng/μl in PBS with 0.1% BSA. Hypothermia-anesthesized pups were injected 2 μl 2xdaily between the femur and tibia, from the external side of the left knee, using a 25 μl syringe (Hamilton).

## Neutralizing antibody treatments

NIMP-R14 (Adipogene #AG-20B-0043PF-C500) was solubilized at 2 mg/ml. 6 µg/g were injected i.p. 2xdaily, using a 25 µl syringe (Hamilton). An equivalent volume of rat IgG or PBS was used as injection control.

## Histology

Mouse pups were euthanized by decapitation after hypothermia-induced analgesia. Knees were dissected out, skinned and fixed by immersion in either 4% paraformaldehyde (PFA, Electron Microscopy Sciences) in PBS for 2 days at 4°C (for immunohistochemistry and in situ hybridization) or 0.25% glutaraldehyde in PBS for 90 min at room temperature (for X-Gal staining). After several washes with PBS, the tissue was then cryoprotected first by brief incubation with a solution of 15% sucrose and then 30% sucrose in PBS for at least 4 hr at 4°C, and then embedded in Cryomatrix (Thermo) using dry-ice-cold isopentane (Sigma). The knees were oriented sagittally and facing each other, with the tibiae on the bottom of the block (i.e. closest to the blade when sectioning). Serial 8-micron sections were collected with a Leica Cryostat on Superfrost slides, allowed to dry for at least 30 min and stored at −80°C until used. For paraffin embedding, the fixative step was followed by 1 week decalcification with EDTA 0.5M in PBS (pH 7.4) at 4°C, dehydration by 30 min incubations with graded ethanol series and xylene at room temperature, and paraffin incubations at 65°C. For all histological techniques, frozen slides were allowed to reach room temperature in a closed box, and Cryomatrix was washed away for 15 min with warm PBS (37°C). Paraffin sections were deparaffinized and rehydrated prior to the staining protocol.

## In situ hybridization

The protocol described in (*Nomura and Hirota, 2003*) was followed. For young pups (P1-P5), samples were not decalcified. Except for *Col2a1, Col10a1* and *Ihh* (provided by Dr. Licia Selleri), the templates for most riboprobes were generated by PCR from embryonic cDNA, using primers containing the SP6 or T7 RNA polymerase promoters. Primer sequences are shown in *Table 1*. After purification of the PCR product (Qiagen PCR purification kit), DIG-labeled probes were transcribed

**Table 1.** Sequence of the primers used to amplify template for riboprobe synthesis from cDNA.

| Primer name | Sequence |
| --- | --- |
| Igf1 F SP6 | GCCGATTTAGGTGACACTATAGAAGTGGATGCTCTTCAGTTCGTG |
| Igf1 R T7 | GAAATTAATACGACTCACTATAGGGTGTTTTGCAGGTTGCTCAAG |
| Fgf18 F SP6 | GCCGATTTAGGTGACACTATAGAAGCCGCCTGCACTTGCCTGTG |
| Fgf18 R T7 | GAAATTAATACGACTCACTATAGGGTGGTTTCTCGCAGTTTCCTC |
| Egr1 F | GTCTTTCAGACATGACAGCGAC |
| Egr1 R SP6 | GCGATTTAGGTGACACTATAGGTGTCACACAAAAGGCACCAA |
| Lef1 F | TGAAGCCTCAACACGAACAG |
| Lef1 R SP6 | GCGATTTAGGTGACACTATAGTTTCCGAAACAACCGTTTTC |
| Hif2a F | CACTGAGACACCTGCCACCTC |
| Hif2a R SP6 | CATTTAGGTGACACTATAGGAGGCACCAGCCACCATG |
| Agc1 F | CCAGCCTGACAACTTCTTTG |
| Agc1 R T7 | GTAATACGACTCACTATAGGGGGGCACATTATGGAAGCTC |
| Fgf18 coding F | GCCGAGGAGAATGTGGACTTCCG |
| Fgf18 coding R SP6 | GCGATTTAGGTGACACTATAGCTAGCCGGGGTGAGTGGGG |
| IL6 F | CTCTGGTCTTCTGGAGTACC |
| IL6 R T7 | CGATGTTAATACGACTCACTATAGGGACCATCTGGCTAGGTAACAG |
| Mcp5 F | GCTTACTCTTCATCTGCTGC |
| Mcp5 R T7 | CGATGTTAATACGACTCACTATAGGGCTGGTGAAGTGTTTGCAGG |

following manufacturer instructions (Roche), treated with DNAase for 30 min and purified by LiCl-mediated precipitation in alcoholic solvent. Probes were kept at $-80°C$ in 50% formamide (Fluka).

## X-Gal staining

For enzymatic detection of $\beta$-galactosidase activity, the frozen sections were postfixed 5 min with 4% paraformaldehyde (PFA, Electron Microscopy Sciences) in PBS at RT. After PBS washes, the sections were incubated $2 \times 5$ min with X-gal buffer (2 mM $MgCl_2$, 0.02% NP40 and 0.05% deoxycholate in PBS 0.1 M pH 7.4) and then overnight at 37°C in X-gal reaction buffer (20 mg/ml X-gal, 5 mM $K_4Fe(CN)_6$ and 5 mM $K_3Fe(CN)_6$ in X-gal wash buffer). After PBS rinses, the sections were postfixed 10 min in 4% PFA and PBS-rinsed again. The sections were then counterstained with Nuclear Fast Red 0.005% for 15 min, serially dehydrated, incubated $3 \times 1$ min with xylene, and cover-slipped using DPX mountant (Fisher).

## Immunohistochemistry and TUNEL

Sections were incubated in citrate buffer (10 mM citric acid, 0.05% Tween 20, pH 6.0) for 15 min at 90°C, allowed to cool down, washed with PBSTx (PBS containing 0.1% Triton X-100), blocked with 5% BSA in PBSTx 30 min at RT, and incubated with the primary antibody o/n at 4°C (see list of antibodies below). After PBSTx washes, incubation with Alexa647- and/or Alexa555-conjugated secondary antibodies (Molecular Probes, 1/500 in PBSTx with DAPI) was performed for 1 hr at RT. After PBSTx washes, the slides were mounted with Fluoro-Gel (Electron Microscopy Sciences). When TUNEL staining was included, it was performed after the citrate step, before the BSA blocking. Endogenous biotin was blocked with the Avidin-Biotin blocking kit (Vector), and the in situ cell death detection kit (Roche) was subsequently used following manufacturer instructions. Biotin-tagged DNA nicks were revealed with Alexa488- or Alexa647-conjugated streptavidin (Molecular Probes, 1/1000) during the incubation with the secondary antibody. The antibodies used were (description, vendor catalog#, dilution): p-S6 (Ser235/236, Cell Signaling #2211S, 1/200), p-STAT3 (Tyr705, Cell Signaling #9145P, 1/200), LEF1 (Cell Signaling #2230P, 1/200), HB-EGF (Diphtheria Toxin Receptor, R and D #AF259NA, 1/300), CD31 (clone MEC 13.3, BD Pharmingen #550274, 1/300), LY6B (clone 7/4, Cedarlane #CL8993AP, 1/100), Collagen Type I (Rockland #600-401-103-0.1, 1/200), OSX (Abcam #ab22552, 1/500), p-FRS2 (Tyr436, R and D #AF5126SP), p-EGFR (Tyr1068, Abcam #ab40815, 1/200), SOCS3 (Abcam #ab16030, 1/100), FABP4 (clone D25B3, Cell Signaling #3544, 1/200), CD44 (clone IM7, BD Pharmingen #550538, 1/500), CD90 (clone G7, eBioscience #14-0901-81, 1/100).

## EdU analysis and quantification of cell parameters

5 mg/ml EdU in PBS was injected subcutaneously (50 μg/g body weight) 1.5 hr before euthanizing the mice. EdU was detected using the Click-iT Alexa488 Imaging Kit (Thermo Fisher Scientific, Waltham, MA, C10337), once the immuno-histochemistry and/or TUNEL staining were finished on the same slides. The fraction of nuclei that were positive for EdU in the proliferative zone of the GP was determined semi-automatically, using Cell Profiler. The number of TUNEL[+] cells in the osteochondral junction and the perichondrial groove of Ranvier were counted manually using Fiji/ImageJ.

## Imaging

Bright-field and fluorescence images were taken on a Zeiss inverted microscope (Observer.Z1) using Axiovision software (Zeiss). Mosaic pictures were automatically reconstructed from individual 10x (brightfield) or 20x (fluorescence) tiles. In some cases, whole slide imaging (WSI) was performed using a Nanozoomer S210 slide scanner (Hamamatsu, Japan).

## Statistics

When data were available for control and experimental animals, a left/right ratio was calculated for both and compared by an unpaired Mann-Whitney test (one variable and two conditions), or by one-way ANOVA (one variable and $\geq 3$ conditions) or by two-way ANOVA (two variables and two or more conditions). When left and right parameters were compared within experimental animals only, a paired two-tailed t-test was used. The data met the assumptions of the tests (e.g. normal distribution by Shapiro-Wilk test). F-test was used to test that the variance was similar between the groups compared. For rescue experiments, animals were assigned to control and experimental groups such

that both groups had similar distributions of initial body weight. For each experiment, the minimum sample size was estimated using an online tool (http://powerandsamplesize.com/Calculators), based on the average SD observed in pilot experiments, to achieve an effect size of 3% (left/right bone length ratio), or 25% (rest of parameters), with a power of 0.8 and a 95% confidence interval. For comparison of qualitative expression, a minimum of two specimens per stage and five across several stages were used. The investigator measuring bone length was blinded to the treatment/genotype of the specimens. No blinding was done for other measurements. Most analyses were done with Prism 7.0 software.

## Tibial culture

A previously described protocol (*Agoston et al., 2007*) was slightly adapted. Briefly, tibiae were obtained from P1 pups, dissected free of soft tissues and allowed to recover from dissection for 6 hr in 24-well plates with serum-free DMEM (Gibco) containing 0.2% Bovine Serum Albumin (BSA), 0.5 mM L-glutamine, 40 U/ml penicillin/streptomycin (Gibco), 0.05 mg/ml ascorbic acid (Sigma) and 1 mM betaglycerophosphate (Sigma). Different quantities of DT in PBS were then added, followed by 24 hr incubation. Tibiae were then fixed in PFA and processed for histological analysis.

## Acknowledgments

We thank the Joyner lab for scientific discussions. We are grateful to Juanma González-Rosa for comments on the manuscript and suggesting the use of *Pitx2-Cre* (kind gift of Hiroshi Hamada), and Chitra Dahia for insightful discussions, critical reading of the manuscript, providing access to the µCT scanner at the Hospital for Special Surgery and for interpretation of the data. This work was supported by grant R21HD083860 (NIH-NICHD) to ALJ, a National Cancer Institute Cancer Center Support Grant [P30 CA008748] to MSKCC, and by postdoctoral fellowships from HFSP and the Revson Foundation to ARD.

## Additional information

### Funding

| Funder | Grant reference number | Author |
| --- | --- | --- |
| Charles Revson Foundation | 15-34 | Alberto Roselló-Díez |
| Human Frontier Science Program | LT000521/2012-L | Alberto Roselló-Díez |
| National Institutes of Health | R21HD083860 | Alexandra L Joyner |

The funders had no role in study design, data collection and interpretation, or the decision to submit the work for publication.

### Author contributions

AR-D, Conceptualization, Formal analysis, Validation, Investigation, Methodology, Writing—original draft, Writing—review and editing; DS, Methodology; ALJ, Supervision, Funding acquisition, Investigation, Project administration, Writing—review and editing

### Author ORCIDs

Alberto Roselló-Díez, http://orcid.org/0000-0002-5550-9846
Alexandra L Joyner, http://orcid.org/0000-0001-7090-9605

### Ethics

Animal experimentation: This study was performed in strict accordance with the recommendations in the Guide for the Care and Use of Laboratory Animals of the National Institutes of Health. All of the animals were handled according to approved institutional animal care and use committee (IACUC) protocols (#07-01-001) of Memorial Sloan Kettering Cancer Center

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
