## [Decision Letter]

Thank you for submitting your article "Altered paracrine signaling from the injured knee joint impairs postnatal long bone growth" for consideration by *eLife*. Your article has been reviewed by two peer reviewers, one of whom, Clifford J Rosen (Reviewer#1) is a member of our Board of Reviewing Editors, and Didier Stainier as the Senior Editor. The following individuals involved in review of your submission have agreed to reveal their identity: Colin Farquharson (Reviewer #2).

The reviewers have discussed the reviews with one another and the Reviewing Editor has drafted this decision to help you prepare a revised submission.

Summary:

In this manuscript, the authors test the hypothesis that local factors outside the growth plate provide signals to long bones to enhance growth during the critical early phases of postnatal development, particularly when growth hormone is not fully activated. In particular, the authors focus on the paracrine activity of the infrapatellar fat pad as a source of IGF-I and activation of the injury response. They propose that Injury and/or inflammation can impact longitudinal growth via disturbed activation of this tissue and suggest that paracrine regulation of the growth plate is essential for normal early neonatal growth. Indeed, understanding the role of these paracrine factors may contribute to our knowledge base about longitudinal growth and pathologic states of impaired growth due to injury.

Essential revisions:

As noted above, the paper breaks new ground with a provocative model system and the rescue experiments. However there were concerns by both reviewers that need to be addressed in the revised manuscript:

1) The authors have not excluded changes in systemic GH/IGF-I axis via the pituitary (particularly since they showed systemic rescue by treating with IGF-I), nor dealt with the possibility in early postnatal development that IGF-II may be altered by inflammation and injury. Delineating changes in these other growth factors is essential for the revision. Furthermore, the discussion of early linear growth is slanted too much toward paracrine factors despite convincing mouse data to the contrary from Arg Efstratiadis and others. Those studies need to be included (see 5 below).

2) Their suggestion that the perichondrium is not the source of altered signaling is not fully supported by the figures and needs clearer delineation and/or further experiments to exclude this source of IGF-I.

3) Further studies examining SOCS3 expression and its role in down regulating IGF-I are necessary as an alternative mechanism.

4) The rescue experiments with ruxo and IGF-I showed lack of a response in the tibia; this needs to be redone or clarified to the extent that it is more demonstrable; we prefer to see a larger N for these experiments.

5) The implications for normal neonatal growth, particularly between days one – twelve is not clear in the discussion and would benefit from further elaboration.

---

## [Author Response]

*Essential revisions:*

*As noted above, the paper breaks new ground with a provocative model system and the rescue experiments. However there were concerns by both reviewers that need to be addressed in the revised manuscript:*

We are glad that the reviewers liked our “provocative model” and we thank them for all their suggestions, as we think the revisions have improved the quality of the manuscript and made the results more compelling.

*1) The authors have not excluded changes in systemic GH/IGF-I axis via the pituitary (particularly since they showed systemic rescue by treating with IGF-I), nor dealt with the possibility in early postnatal development that IGF-II may be altered by inflammation and injury. Delineating changes in these other growth factors is essential for the revision. Furthermore, the discussion of early linear growth is slanted too much toward paracrine factors despite convincing mouse data to the contrary from Arg Efstratiadis and others. Those studies need to be included (see 5 below).*

This is indeed a very important point that was not completely clear in the original manuscript. Since we are performing a unilateral modification, with the right limb as internal control, any left-specific defect cannot be due to changes in a systemic factor(s) (be it GH or IGF1), although it could be due to local changes in the signal transduction machinery that responds to those systemic factors. Importantly, Efstratiadis and others have shown that mutant mice for *growth hormone receptor (Ghr*) do not show a body weight or a limb length phenotype until P10-P15 (Zhou et al., PNAS 1997, Lupu et al., Dev Biol 2001), while we see a significant effect as early as P4. Therefore, we decided to focus on local IGF signaling. We did not detect major changes of expression for *Igf2* or *Igf1r* (i.e. the other IGF ligand and their common receptor), except in the damaged cells of the prospective articular cartilage, where the *Igf2* expression domain is somewhat interrupted (new Figure 4—figure supplement 1) To test the possibility that downstream signaling components were affected, we determined SOCS3 expression as suggested by the reviewer, as it could conceivably be activated by inflammatory cytokines and exert a negative influence on IGF signaling (Ahmed and Farquharson, J. Endocrinology 2010). However, we could not detect differential expression between the left and right GPs in P1-*PitDTR* animals at P2 or P3 (new Figure 4—figure supplement 1), suggesting that it is the reduced availability of IGF1 from the fat pad that plays a major role in the phenotype. It is also noteworthy that when the same intra-articular (i.e. local) IGF1 injection regimen that we performed in the rescue experiment is reproduced in *PitDTR* animals not given DT, no overgrowth phenotype is observed (Figure 8). Actually the injected limb is slightly shorter, perhaps due to the repeated injections.

Author response image 1.Left and right bone length in P5 *PitDTR* animals that were not injected with DT, but that were injected with IGF1 in the left knee from P1 to P4, twice daily. p-values for multiple comparisons test after two-way ANOVA are shown.**DOI:**
http://dx.doi.org/10.7554/eLife.27210.022

We have also expanded the Introduction/Discussion to further clarify the role of IGF signaling in early postnatal development. We explain that deletion of *Igf1* in the liver (which diminishes circulating levels by >75%) does not affect limb growth (Yakar et al.,and Sjögren et al., PNAS 1999), whereas chronic overexpression of *Igf1* in liver, brain and other organs results in overgrowth of only a subset of organs “without an apparent increase in skeletal growth” (Mathews et al., Endocrinology 1988). Furthermore, one of the most comprehensive studies relating to the trophic role of GH/IGF, by the Efstratiadis lab (Stratikopoulos et al., PNAS 2008), reached the conclusion that “growth hormone-induced, liver-specific IGF1 apparently begins exerting postnatal hormonal action shortly after weaning”. Finally, the fact that deletion of *Igf1* in the whole limb mesenchyme greatly diminishes chondrocyte hypertrophy by P7 (Cooper et al., Nature 2013), whereas specific *Igf1* deletion in chondrocytes from embryonic stages onwards does not affect bone growth by two weeks of age (Govoni et al., Physiol Genomics 2007), strongly argues that the main local source.

*2) Their suggestion that the perichondrium is not the source of altered signaling is not fully supported by the figures and needs clearer delineation and/or further experiments to exclude this source of IGF-I.*

In the original manuscript, we chose to show *Igf1* expression only in the IPFP because it is the only tissue in which we detected diminished *Igf1* levels. We agree with the reviewer that showing other source tissues makes the result more compelling, and therefore we have included close-ups of *Igf1* expression in several regions of the limb mesenchyme (Figure 4—figure supplement 1). As shown in the figure, *Igf1* expression is not changed in the left perichondrium, metaphyseal region or muscle bundles surrounding the bones, as compared to the right limb. It should be noted that perichondrial *Igf1* expression is only detected at certain section levels, and due to inevitable differences in the precise orientation of each bone during sectioning, it is not possible to obtain exactly matched section levels for left and right bones. Therefore, to assess changes of expression as thoroughly as possible, we decided to image every single section from 5 different animals using a slide scanner. Through this serial analysis, we only found consistent left-right differences in *Igf1* expression in the IPFP.

*3) Further studies examining SOCS3 expression and its role in down regulating IGF-I are necessary as an alternative mechanism.*

Please see our response to point 1 for our new results on SOCS3 expression.

*4) The rescue experiments with ruxo and IGF-I showed lack of a response in the tibia; this needs to be redone or clarified to the extent that it is more demonstrable; we prefer to see a larger N for these experiments.*

We have now increased the number of rescued animals (especially for the double rescue, from six to eleven), and we have modified the statistical analysis, such that instead of analyzing femur and tibia separately by one-way ANOVA with Treatment as variable, we performed twoy ANOVA with Bone and Treatment as variables, in order to take into account that the measurements of femur and tibia from the same animal are matched and not independent. Interestingly, only the p-value for Treatment is statistically significant, meaning that there is not a consistent difference between femur and tibia across treatments. The conclusion of the posthoc test for multiple comparisons is that both bones show a trend towards rescue, with the p-value being 0.047 for the femur and 0.059 for the tibia (see new Figure 6). As thresholds for statistical significance are arbitrary, another way to assess the biological relevance is to consider that 3/11 femora were completely rescued, whereas none of the tibiae showed complete rescue. Therefore, we cannot discard that the response of femur and tibia to the same growth-modulating cues varies slightly depending on the time window during which the perturbation happens. Indeed, the specific effect of limb immobilization on the growth of different bones was recently shown to depend on the time of treatment in chicken and crocodile embryos (Pollard et al., Sci Rep 2017). We have included this possibility in the discussion.

*5) The implications for normal neonatal growth, particularly between days one – twelve is not clear in the discussion and would benefit from further elaboration.*

We agree that this is an important point, and while in the original manuscript we briefly discussed the potential role of the fat pad in the establishment of body proportions and allometric growth during the normal early postnatal period, we have now expanded the discussion to make this implication more clear. We have also added a developmental description of fat pad formation and the temporal correlation between *Igf1* expression in this tissue and the period in which the hindlimb grows faster than the forelimb in mice (see Figure 7—figure supplement 1). In addition, we have included in the Discussion the possibility that the fat pad, and/or other local tissues hosting cells with osteochondrogenic potential, such as the synovium, bone marrow or the perichondrial groove of Ranvier (Yang et al., Nature 2013, Karlsson et al., J. of Anatomy 2009, Hindle et al., Stem Cells Transl Med 2017; Chung and Xian, J. Mol. Endocrinology 2014), do not only contribute with paracrine signals to bone growth, but also with progenitor cells*, such that cell death within these tissues in *PitDTR* mice depletes the pool of cells that could potentially participate in recovery of the affected bones. * (see for example Fig. S9 in Yang et al. Nature 2013)